# The awareness of chronic kidney disease in non-medical female university students: A cross-sectional in Riyadh, Saudi Arabia

**Amen Bawazir[ID], Maria Fayoumi, Alaa Alharbi, Arwa Alakeel, Bashayer Alskait, Areej Alsanawi, Manar Alahmari, Hala Alamer, Kholoud Alsaygh, Thekra Khattar, Njood Aleid, Mohammad Fareed[ID]***

Community Medicine Unit, Department of Basic Medical Sciences, College of Medicine, AlMaarefa University, Diriyah, Riyadh, Saudi Arabia

* fareed.research@gmail.com

## Abstract

### Background

Assessing the level of knowledge about Chronic Kidney Disease (CKD) is very crucial for society. Research on CKD knowledge can help to identify gaps in public understanding and develop targeted educational interventions to improve public awareness of CKD.

### Aim

To assess the level of awareness of CKD among non-medical students in Riyadh region of Saudi Arabia.

### Methods

A cross-sectional study was conducted between February 2023 to November 2023 including 523 non-medical female university students in Riyadh region. A self-administer questionnaire was designed and used for collecting the required data. The information viz. age, nationality, residential area, university, educational level, study field, academic year, and income were obtained from each participant. Information for knowledge of CKD knowledge and associated risk factors were also obtained in the study.

### Results

The study concluded non-medical female university students had an unsatisfactory level of knowledge regarding CKD. We found a significant association ($p < 0.05$) between the adequacy of general knowledge, knowledge of risk factors, complications, and active preventive measures regarding CKD with the sociodemographic characteristics of the participants.

**Data availability statement:** All relevant data are within the article and its supporting information files.

**Funding:** The author(s) received no specific funding for this work.

**Competing interests:** The authors have declared that no competing interests exist.

## Conclusion

The study found that most non-medical female students have an unsatisfactory level of knowledge regarding CKD, therefore, educational materials should be developed and distributed to inform people more about CKD risk factors, symptoms, and treatment options like organizations that can offer educational events and webinars on CKD. These events can be a great way to learn about CKD from experts in the field and to ask questions about specific concerns.

## Introduction

According to National Institute of Diabetes and Digestive and Kidney Diseases (NIDDK), kidney failure is a medical illness in which the kidneys are no longer able to function at levels that are sufficient to filter the blood's waste products. It's also referred to as chronic kidney disease (CKD) or end-stage renal disease (ESRD) [1]. Acute renal failure (ARF) which manifests quickly and may resolve, is distinguished from chronic renal failure (CRF), which manifest gradually and frequently becomes irreversible [2].

CKD is a growing global health concern that significantly affects morbidity and mortality rates worldwide [3]. The prevalence of CKD has steadily increased over the past few decades due to the rising incidence of risk factors such as diabetes, hypertension, and obesity [4]. In Saudi Arabia, CKD poses a particular challenge, with the country witnessing a steady rise in the number of patients suffering from end-stage renal disease (ESRD) [5]. Despite the severity of CKD and its associated complications, public awareness regarding the disease, especially in non-medical populations, remains insufficient [6]. Studies have shown that early detection and lifestyle modifications can significantly delay the progression of CKD and improve patient outcomes [7]. However, limited research has been conducted on the awareness and knowledge of CKD among young adults, particularly among university students who are not pursuing medical degrees [8].

In Saudi Arabia, the level of CKD knowledge among female university students is particularly important, as women often play a central role in health education within families [9]. In Saudi Arabia, women serve as central figures in family health education, guiding practices related to hygiene, nutrition, and disease prevention. Cultural and religious values position them as key caregivers, with their influence further supported by national goals like Vision 2030. Their role became especially evident during the COVID-19 pandemic, as they enforced safety measures and helped manage household responses. Studies recognize Saudi women as the "first line of defense" in protecting families and promoting public health awareness [10].

Understanding the awareness levels in this demographic group could have far-reaching implications for public health initiatives aimed at CKD prevention and early intervention [11].

Targeting non-medical female university students to assess Chronic Kidney Disease (CKD) awareness is strategic for identifying general knowledge gaps,

addressing gender-specific disparities and risks, and leveraging women's roles in health promotion. Moreover, it is justified by their unique hormonal and reproductive risk factors for CKD, the opportunity to assess health behaviors and promote health literacy, their accessibility as a research population, and the chance to address potential CKD knowledge gaps and misconceptions. This approach facilitates the development of targeted educational interventions to enhance early detection and prevention efforts. Similar study with the same target group was conducted among non-medical students in the University of Nigeria [12], Rwanda [13], and USA [14].

Previous studies that discuss the awareness of CKD in Saudi Arabia showed that the general Saudi population does not have enough level of awareness about CKD. Therefore, this study aims to assess the level of awareness and knowledge of CKD among non-medical female university students in Riyadh, Saudi Arabia, and identify potential gaps in knowledge that could be addressed through targeted educational programs.

## Methodology

This study employed a descriptive cross-sectional survey design targeting non-medical female university students in Riyadh, Saudi Arabia. Ethical approval was obtained from the Institutional Review Board (IRB) at AlMaarefa University (Approval No. IRB23–041). Prior to participation, verbal consent was obtained from all students after explaining the study's purpose and ensuring voluntary participation. The study included female students from non-medical disciplines Female medical students and all male students were excluded.

The sample size was determined using a prevalence rate of 27.8% for adequate knowledge of chronic kidney disease (CKD), based on a previous study from Jazan [15]. Using the standard formula with $Z = 1.96$, $P = 0.278$, $q = 0.65$, a 95% confidence level, and a 5% margin of error, an initial sample of 305 participants was calculated. This was adjusted for a design effect of 1.5, resulting in a final sample size of 523.

A multistage sampling technique was applied. Across the 16 universities in Riyadh (comprising 11 private and 5 public institutions) four universities were randomly chosen in the study (2 publics and 2 private) [16,17]. Participants were proportionally distributed based on the student population in each selected institution. For public universities, King Saud University and Imam Mohammad Ibn Saud Islamic University, students were selected with a total of 334 participants equally split between the two institutions. For private universities, Prince Sultan University and Al-Yamamah University, students were selected with a total of 166 participants equally distributed between both universities.

## Data collection and measures used

A structured, self-administered questionnaire—designed based on validated instruments from prior studies [14,16–18], was distributed as printed copies to target participants across multiple university campuses. This in-person approach ensured direct participant engagement, comprehensive responses, and efficient data collection. Data collection began on 5th February to 21st March, 2023, by a team of trained students from the College of Medicine at AlMaarefa University, who followed standardized protocols for accuracy and consistency. The questionnaire was first piloted among 5% of the sample (35 students), and then reviewed and assessed by 3 subject experts for its content, design, relevance, readability and comprehension. A content validity ration (CVR) was calculated for each domain and all domains reported 0.75 for knowledge domain, 0.70 for attitude domain, and 0.76 for practice domain for Cronbach alpha coefficients score. Domains with less than 0.07 were reviewed carefully and targeted questions were modified or deleted accordingly. Three questions related to the knowledge from the piloted version were found not appropriate and then not included in the final version of the questionnaire.

The final version of the questionnaire comprised 53 questions, structured into six main sections: sociodemographic characteristics (7 questions), knowledge of CKD (12 questions), CKD risk factors related questions (11 questions), CKD complication related questions (7 questions), questions related to Active prevention of CKD (8 questions), and questions related to Source of information on chronic kidney disease (8 questions). Some questions were scored with a binary system, where incorrect or uncertain ("don't know") responses received a score of 0, while correct answers were awarded

1 point. The questions related to CKD risk factors, CKD complication, q and questions related to Active prevention utilized a five-point Likert scale (strongly disagree to strongly agree), with responses scored from 1 (least appropriate) to 5 (most appropriate). To account for unfavorable statements, scores for such items were inverted during analysis. These domains (CKD risk factors, CKD complication, q and questions related to Active prevention) had a possible score range of 11–55, 7–35, and 8–40, respectively. Criteria for correct/incorrect answers were determined through prior literature and expert validation during questionnaire development. This scoring framework ensured consistency in evaluating participants' related status, which aligning with established methodologies to enhance reliability and comparability of results.

Variables related to the general knowledge of the Kidneys and CKD, an example of the questions: CKD is a common disease. Variables related to the awareness of CKD's risk factors such as diabetes and hypertension are risk factors for CKD. Variables related to the complications related to CKD, for example, anemia is a complication for CKD. Variables related to the active preventive measures related to CKD, for example: practicing physical activities can prevent kidney disease. Moreover, some variables related to the source of information on CKD, just like your main source of information regarding CKD is family, were included in the questionnaire.

The overall knowledge score was then computed by the sum of all answers and a percentage was established to recognize the percentages of those having positive or negative knowledge about risk factors, complications, and active preventive measures associated with CKD, by the use of the mean value as a cut-off point in this measurement.

## Data analysis

To ensure the quality and accuracy of the data, all the questionnaires were reviewed before entering their data into the SPSS (Statistics Program for Social Sciences, Windows version 25.0 IBM Corp., Armonk NY, USA) as the analysis program used in this study. Missing data were handled by excluding incomplete responses; only fully completed question-naires were included in the final analysis to ensure data accuracy and consistency.

Results were expressed accordingly as number (%), mean, standard deviation (SD), or median (min-max). Pearson's chi-square analysis was performed for categorical variables, and the Shapiro-Wilk tests were used to test assumptions of normality. Spearman's test was used to analyze the correlation between ordinal variables (academic year and other socio-demographic data) and knowledge of CKD, ensuring multicollinearity checks prior to regression analysis.

For determining the level of knowledge of the participants, an association test was used between the adequacy of general knowledge, knowledge of risk factors, complications, and active preventive measures regarding CKD with the sociodemographic characteristics of the participants using a p-value of less than 0.05 as a significant value for interpreting the findings.

## Results

### Sociodemographic characteristics of the participants

Out of the total number of participants (523), the median age of the participants was 20 years (IQR: 17–30 years). The majority of the participants were aged (20–21) years (38.8%), Saudi (91.4%), Northern area of Riyadh (40.2%), from Imam Mohammed Ibn Saud University (33.3%), from the science college (44.6%), of the fourth academic year (26.0%), and those with a monthly income of more than 10,000 Saudi Riyal (SAR) (61.2%) as shown in Table 1.

### Association of different sociodemographic characteristics with knowledge of CKD

Table 2 exhibits the association of different sociodemographic characteristics with knowledge of CKD among students. We found that the variable of universities and academic year were significantly associated (p < 0.05) with different aspects of knowledge among students, however other variables do not show significant association. The study found that students' knowledge of CKD—across general information, risk factors, complications, and preventive measures—was significantly

**Table 1. Sociodemographic characteristics of the participants.**

| Variables | Categories | N | % |
|---|---|---|---|
| Age groups | ≤19 years | 189 | 36.1 |
| | 20-21years | 203 | 38.8 |
| | ≥22 years | 131 | 25.0 |
| Nationality | Saudi | 478 | 91.4 |
| | Non-Saudi | 45 | 8.6 |
| Residential Area | Central | 103 | 19.7 |
| | Southern | 41 | 7.8 |
| | Northern | 210 | 40.2 |
| | Western | 49 | 9.4 |
| | Eastern | 120 | 22.9 |
| University | King Saud university | 165 | 31.5 |
| | Imam Mohammed Ibn Saud | 174 | 33.3 |
| | Prince Sultan University | 101 | 19.3 |
| | Al-Yamamah University | 83 | 15.9 |
| Educational program | Bachelor | 514 | 98.3 |
| | Diploma | 9 | 1.7 |
| Study Field | Humanitarian | 160 | 30.6 |
| | science college | 233 | 44.6 |
| | business college | 130 | 24.9 |
| Academic year | Preparatory Year | 89 | 17.0 |
| | first year | 110 | 21.0 |
| | second year | 100 | 19.1 |
| | third year | 88 | 16.8 |
| | fourth year | 136 | 26.0 |
| Income | < 5000 SR | 66 | 12.6 |
| | 5000-10,000 SR | 137 | 26.2 |
| | > 10,000 SR | 320 | 61.2 |

associated with their university and academic year. Prevention knowledge was higher in King Saud and Prince Sultan students compared to Al-Yamamah. Educational context drives awareness more than other demographics. Other sociodemographic factors did not show significant associations, indicating that institutional and academic-level differences play a key role in shaping CKD-related knowledge.

**Regression Model analysis for the knowledge of CKD preventive practice and sociodemographic characteristics**

Table 3 presents the results of a regression model examining the relationship between the knowledge of CKD preventive practices and various sociodemographic characteristics. The main findings suggest that students from King Saud (AOR: 2.005; 95% CI: 1.173–3.426; P-value: 0.011), and Prince Sultan universities (AOR: 1.867, 95%CI: 1.037–3.362; p-value: 0.038) are more likely to have knowledge about CKD compared to Al-Yamamah University. However, the other sociodemographic variables do not appear to have a significant association with CKD knowledge based on this analysis.

## Discussion

The current study evaluated the general awareness of CKD among female non-medical university students in Riyadh. The findings of this study show that female university students in Riyadh, Saudi Arabia, who are not medical professionals

**Table 2. Association of different sociodemographic characteristics with knowledge of CKD.**

| Variables | Categories | General Knowledge of CKD | | | Knowledge for Risk Factors of CKD | | | Knowledge for Complications of CKD | | | Knowledge for Prevention Practice of CKD | | |
|---|---|---|---|---|---|---|---|---|---|---|---|---|---|
| | | N | % | P value | N | % | P value | N | % | P value | N | % | P value |
| Age groups | ≤19 years | 86 | 45.5 | .857 | 89 | 47.1 | .427 | 98 | 51.9 | .446 | 103 | 54.5 | .734 |
| | 20-21years | 98 | 48.3 | | 85 | 41.9 | | 93 | 45.8 | | 116 | 57.1 | |
| | ≥22 years | 61 | 46.6 | | 53 | 40.5 | | 61 | 46.6 | | 77 | 58.8 | |
| Nationality | Saudi | 224 | 46.9 | .980 | 208 | 43.5 | .867 | 229 | 47.9 | .681 | 266 | 55.6 | .154 |
| | Non-Saudi | 21 | 46.7 | | 19 | 42.2 | | 23 | 51.1 | | 30 | 66.7 | |
| Residential Area | Central | 46 | 44.7 | .540 | 38 | 36.9 | .309 | 49 | 47.6 | .979 | 63 | 61.2 | .730 |
| | Southern | 15 | 36.6 | | 18 | 43.9 | | 21 | 51.2 | | 21 | 51.2 | |
| | Northern | 104 | 49.5 | | 100 | 47.6 | | 101 | 48.1 | | 120 | 57.1 | |
| | Western | 21 | 42.9 | | 24 | 49.0 | | 25 | 51.0 | | 25 | 51.0 | |
| | Eastern | 59 | 49.2 | | 47 | 39.2 | | 56 | 46.7 | | 67 | 55.8 | |
| University | King Saud | 69 | 41.8 | .068 | 75 | 45.5 | .466 | 80 | 48.5 | .796 | 93 | 56.4 | .025 |
| | Al-Imam | 83 | 47.7 | | 69 | 39.7 | | 85 | 48.9 | | 86 | 49.4 | |
| | Prince Sultan | 44 | 43.6 | | 49 | 48.5 | | 51 | 50.5 | | 69 | 68.3 | |
| | Al-Yamamah | 49 | 59.0 | | 34 | 41.0 | | 36 | 43.4 | | 48 | 57.8 | |
| Educational level | Bachelor | 243 | 47.3 | .135 | 222 | 43.2 | .458 | 248 | 48.2 | .821a | 291 | 56.6 | .949 |
| | Diploma | 2 | 22.2 | | 5 | 55.6 | | 4 | 44.4 | | 5 | 55.6 | |
| Study Field | Humanitarian | 72 | 45.0 | .355 | 73 | 45.6 | .117 | 78 | 48.8 | .983 | 87 | 54.4 | .157 |
| | science college | 105 | 45.1 | | 90 | 38.6 | | 112 | 48.1 | | 126 | 54.1 | |
| | business college | 68 | 52.3 | | 64 | 49.2 | | 62 | 47.7 | | 83 | 63.8 | |
| Academic year | Preparatory | 43 | 48.3 | .285 | 43 | 48.3 | .687 | 46 | 51.7 | .895 | 49 | 55.1 | .022 |
| | first year | 53 | 48.2 | | 44 | 40.0 | | 51 | 46.4 | | 49 | 44.5 | |
| | second year | 37 | 37.0 | | 47 | 47.0 | | 50 | 50.0 | | 66 | 66.0 | |
| | third year | 43 | 48.9 | | 36 | 40.9 | | 43 | 48.9 | | 55 | 62.5 | |
| | fourth year | 69 | 50.7 | | 57 | 41.9 | | 62 | 45.6 | | 77 | 56.6 | |

**Table 3. Regression model for the knowledge of CKD preventive practice and sociodemographic characteristics.**

| Variables | Categories | OR | 95% CI | P value | AOR | 95% CI | P value |
|---|---|---|---|---|---|---|---|
| Age groups | ≤19 years | 1.031 | .508-2.092 | .932 | – | – | – |
| | 20-21years | .843 | .502-1.418 | .520 | – | – | – |
| | ≥22 years | R | – | – | – | – | – |
| Nationality | Saudi | .970 | .486-1.935 | .931 | – | – | – |
| | Non-Saudi | R | – | – | – | – | – |
| Educational program | Bachelor | .371 | .072-1.922 | .237 | – | – | – |
| | Diploma | R | – | – | – | – | – |
| Study Field | Humanitarian | 1.212 | .731-2.009 | .457 | – | – | – |
| | science college | 1.279 | .809-2.023 | .292 | – | – | – |
| | business college | R | – | – | – | – | – |
| University | King Saud | 2.098 | 1.152-3.822 | **.015** | 2.005 | 1.173-3.426 | **.011** |
| | Imam Mohammed | 1.518 | .821-2.806 | .183 | 1.580 | .931-2.682 | .090 |
| | Prince Sultan | 1.761 | .937-3.309 | .079 | 1.867 | 1.037-3.362 | **.038** |
| | Al-Yamamah | R | – | – | R | – | – |
| Income | < 5000 SR | 1.213 | .690-2.131 | .502 | – | – | – |
| | 5000-10,000 SR | 1.202 | .778-1.857 | .407 | – | – | – |
| | > 10,000 SR | R | – | – | – | – | – |

have unsatisfactory knowledge level regarding CKD's general knowledge, where only 53.2% had adequate knowledge. Our findings fell a little bit short of a study done in Saudi Arabia [15,18] where 57.1% had adequate knowledge. there could be a reason for the variations in knowledge levels between our study and their study. One hypothesis is that compared to the university students in the current study, the participants in their study came from a wider variety of backgrounds.

Moreover, our findings suggest that there is no significant association between sociodemographic characteristics and knowledge of CKD, but some variations were found, and we think that the following hypotheses may explain theses variations. For example, we think that our unsatisfactory level of knowledge might be related to our sample age group, since more than half of our participants were young adults and CKD is not common in this age.

In the present study, one of the most significant findings was that the knowledge about CKD had no relation to income, where Participants with the highest income had the lowest knowledge levels 51.2%, we think that this may be attributed to the better healthcare access and screening among well-off population, on the contrary the Saudi study [15] found a link between higher levels of knowledge and higher economic position. Furthermore, our study demonstrated a positive correlation between the adequate level of knowledge among southern residents in Riyadh 63.4%, this might be related to the presence of Al Iman General Hospital and its health awareness activities in that area.

Furthermore, our study also found that participants majoring in science college had better levels of knowledge about CKD than those majoring in business. This is likely because science majors are more likely to take courses in biology and anatomy, which teach about the kidneys and their functions.

Our study found that only 56.6% of non-medical female university students in Riyadh City, Saudi Arabia, demonstrated adequate knowledge of CKD risk factors, which is higher than a previous Saudi study [13] where only 31.6% of participants had moderate to high knowledge. This difference may be attributed to our sample consisting solely of university students, while their [13] sample included students, employed, and unemployed individuals, some of whom had only completed primary school.

Furthermore, our results indicate the absence of a notable correlation between sociodemographic traits and awareness of CKD risk factors. However, certain deviations were identified, and we posit that the following hypotheses may elucidate these variations. The highest level of awareness was found among the older age group (≥22 years) 59.5%, which is understandable given that they could be exposed to more information with age. Moreover, the highest level was also found in participants living in the center of Riyadh 63.1%, this could be due to the presence of multiple hospitals in that area which can perform health awareness campaigns. One thing was surprising, that the highest level was found among participants with an average family income of 59.1% which contraindicates the findings of previous studies stating that a higher level of awareness is related to higher income levels [15,12,19–21].

The higher knowledge levels of CKD among students at King Saud University (KSU) can be linked to its competitive admission standards, which attract academically strong students [22]. The university emphasizes quality teaching, supported by experienced faculty and multimedia tools, creating an engaging learning environment [23]. Additionally, its wide range of extracurricular activities provides students with learning opportunities beyond the classroom, further enhancing their knowledge and academic growth [24]. The lowest level of awareness was found in participants studying in Prince Sultan University 51.5%, the way we can explain this finding is that Prince Sultan University does not have any health-related college, thus limiting the students' exposure to health-related information.

Our study described the level of awareness of CKD complications among non-medical female university students in Riyadh city. Saudi Arabia. The findings demonstrate that our target group has an unsatisfactory level of awareness (51.8%). Although our findings were unsatisfactory, it was higher than two studies both done in Saudi Arabia, one was done in the Hail region [11] and had 43.5% of their participants with adequate knowledge, and another one done in the AlHassa region [25] with 36% with adequate knowledge.

We hypothesize that our results differ from the Hail study [11] due to our larger sample and more specific target group: non-medical female university students in Riyadh City, versus the general Hail population. We found no significant

relationship between CKD complications knowledge and sociodemographic characteristics, though some variables showed more effect."

Our findings demonstrate that the highest level of knowledge regarding CKD complications was seen in participants in their fourth academic year 54.4% having family monthly income of >10,000 SR (54.7%). We can explain these findings by saying that the older the participants the more knowledge and exposure to health-related information they get, furthermore, the more income they have the more access to better healthcare they have. The lowest level of knowledge was seen among participants studying in humanitarian colleges 51.2%, which could be explained by the very little exposure to health-related information they have. Our findings correspond to the findings of a Saudi study [15], where they concluded that higher levels of knowledge were associated with higher income levels.

This study revealed low awareness of chronic kidney disease (CKD) preventive practices (43.4%) among non-medical female university students in Riyadh, Saudi Arabia. In contrast, a Brazilian study [26] reported significantly higher knowledge levels (91.78%) regarding CKD prevention. The disparity in findings may reflect differences in sample size, with the Brazilian study involving a larger participant cohort.

When associating the level of knowledge with the sociodemographic characteristics of the participants the only factors that were significantly associated were the university attended and the academic year the participants are studying in (p-value (>0.05)). furthermore, no other factor had a significant association.

It is suggested that the reason for this significant association is that some universities may have more educational programs than others, especially if that university has a medical college. Moreover, the reason for the significance of the academic year may be due to more access of some age groups to social media that present large quantity of information, or because some batches could conduct awareness campaigns.

One extremely surprising finding is that the highest level of knowledge was seen among the participants with the lowest family monthly income 48.5%. One way to explain this is that people with low income could follow traditional ways of treatment or could have the traditional correct information about preventing diseases.

Moreover, another surprising finding was that the highest level of knowledge was seen among the youngest participants, 45.5%. We can explain this by the hypothesis that the younger the generation is, the more it is exposed to social media, in addition to the fact that the younger the people are the more resilient they are to obey instructions especially when it comes to their health. Digital campaigns and social media offer great promise for raising Chronic Kidney Disease (CKD) awareness among younger, who frequently use these tools for information. Consistent with the idea that younger generations turn to social media, websites, and health apps for knowledge, these tools enable targeted health education. Through interactive features, infographics, brief videos, and influencer-driven efforts, they can break down intricate medical details, making CKD risks, symptoms, and prevention strategies more approachable and compelling.

Targeted health education programs can boost CKD awareness among non-medical female students at Saudi Universities. Awareness campaigns on popular Saudi social media like Twitter and Instagram, using hashtags like "#KnowYourKidneys" with engaging content on risk factors (e.g., poor diet, inactivity) and prevention (e.g., hydration, exercise), aligned with events like World Kidney Day [1]. Interactive workshops led by medical professionals, offering screenings and tailored discussions [2], and integrating CKD topics into general university courses [3], can provide foundational knowledge. Peer-led initiatives like student-hosted "Kidney Talks" can further engage students informally [4]. These strategies address awareness gaps effectively.

## Strength and limitation

### Strengths

To our knowledge, very less researches were conducted to assess and explore the awareness of CKD among non-medical female university students in Riyadh city, Saudi Arabia. Moreover, our sample size was bigger than some other research done in Saudi Arabia.

## Limitations

This was a cross-sectional study which may cause some information bias. Furthermore, our target group was only in one city in Saudi Arabia, this could have caused bias.

## Conclusion

The study concludes that the level of awareness of CKD among non-medical female university students in Riyadh city was moderate, moreover, we did not find any significant relationship between the level of knowledge and the participants' sociodemographic information, except for the information related to the knowledge of active preventive measures of CKD where there was a significant influence of the university and the academic year on the level of knowledge. Furthermore, the lowest levels of knowledge were found in the information regarding preventive practices.

We recommend that more health educational programs should be implemented, especially among populations of young age to increase the chance of disease prevention.

## Supporting information

**S1 Raw Data. The unalysed raw data for all the applicable variables of the research.**
(XLSX)

## Acknowledgments

The authors express their thanks and gratitude to AlMaarefa University, Riyadh, Saudi Arabia for the support to publish this article. Additionally, authors are thankful to Dr. Ali Assiry for permitting us to use his validated questionnaire.

## Author contributions

**Conceptualization:** Amen Bawazir, Thekra Khattar.

**Data curation:** Amen Bawazir, Alaa Alharbi, Areej Alsanawi, Arwa Alakeel, Bashayer Alskait, Kholoud Alsaygh, Manar Alahmari, Maria Fayoumi, Njood Aleid, Ragedah Alamer.

**Formal analysis:** Amen Bawazir.

**Investigation:** Amen Bawazir, Alaa Alharbi, Areej Alsanawi, Arwa Alakeel, Bashayer Alskait, Kholoud Alsaygh, Manar Alahmari, Maria Fayoumi, Njood Aleid, Ragedah Alamer, Thekra Khattar.

**Methodology:** Amen Bawazir, Alaa Alharbi, Areej Alsanawi, Arwa Alakeel, Bashayer Alskait, Kholoud Alsaygh, Manar Alahmari, Maria Fayoumi, Njood Aleid, Ragedah Alamer, Thekra Khattar.

**Project administration:** Amen Bawazir.

**Resources:** Amen Bawazir.

**Software:** Amen Bawazir.

**Supervision:** Amen Bawazir.

**Validation:** Amen Bawazir, Mohammad Fareed.

**Visualization:** Amen Bawazir, Mohammad Fareed.

**Writing – original draft:** Amen Bawazir, Alaa Alharbi, Areej Alsanawi, Arwa Alakeel, Bashayer Alskait, Kholoud Alsaygh, Manar Alahmari, Maria Fayoumi, Njood Aleid, Ragedah Alamer, Thekra Khattar.

**Writing – review & editing:** Amen Bawazir, Mohammad Fareed.

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
