## [Decision Letter · Decision Letter 0]

28 Feb 2025

PONE-D-24-49522The Awareness of Chronic Kidney Disease in Non-Medical Female University Students: A Cross-Sectional in Riyadh, Saudi ArabiaPLOS ONE

Dear Dr. Fareed,

Thank you for submitting your manuscript to PLOS ONE. After careful consideration, we feel that it has merit but does not fully meet PLOS ONE’s publication criteria as it currently stands. Therefore, we invite you to submit a revised version of the manuscript that addresses the points raised during the review process.

We look forward to receiving your revised manuscript.

Kind regards,

Jordan Llego, PhD ELM, D. Hon. Ex., PhDN, RN

Academic Editor

PLOS ONE

Additional Editor Comments:

The study addresses an important and underexplored public health issue, but revisions are necessary to elevate its impact. The manuscript could benefit from several enhancements to improve its credibility and clarity.

Firstly, it is essential to strengthen justifications and citations throughout the text. For example, supporting citations are needed to validate the claims made in discussing women's roles in health education. Additionally, the introduction must explicitly explain the rationale for focusing on non-medical female university students, detailing how this demographic uniquely contributes to Chronic Kidney Disease (CKD) awareness efforts. Incorporating references to prior CKD awareness studies in similar populations would further solidify the foundation of the research.

Furthermore, the methodology needs greater transparency. When discussing the questionnaire, whether it was adapted from validated instruments, piloted, or verified for reliability should be clear. The manuscript should also specify how missing data were handled, indicating if incomplete responses were excluded or imputation was used. More information on participant recruitment is also necessary, particularly regarding how the universities were chosen and whether any stratification or weighting was applied in the analysis.

Statistical and data presentation can be refined for better clarity. For instance, Table 1 should include percentage values alongside raw numbers to enhance understanding. Significant statistical associations need to feature confidence intervals and p-values for improved interpretation. If regression models were employed, it would be prudent to state whether multicollinearity diagnostics were conducted. Additionally, statistically significant p-values should be highlighted in bold for easy reference, and figures and tables should be high-resolution with clear labels.

The discussion section requires enhancements to interpret and summarize key findings effectively. Each knowledge domain—general awareness, risk factors, complications, and preventive measures—should be succinctly summarized in the text. The manuscript should also address the disparities in CKD knowledge, notably why certain universities, like King Saud University, demonstrated higher awareness levels. Were specific health programs or university initiatives influential in this regard? Moreover, considering the young demographic, exploring the impact of digital campaigns and social media on CKD awareness would add valuable depth to the discussion. Instead of an overly general call for awareness programs, specifying practical interventions, such as university-integrated CKD education or student-led health initiatives, would enhance the applicability of the findings.

Lastly, attention should be paid to language, structure, and formatting. Several sentences are overly lengthy and could be revised for better readability. Consistency in terminology—specifically, the terms "CKD knowledge" and "CKD awareness"—should be maintained throughout. Finally, the reference formatting must adhere to PLOS ONE style guidelines to ensure proper citation structure.

Reviewers' comments:

Reviewer's Responses to Questions

**Comments to the Author**

1. Is the manuscript technically sound, and do the data support the conclusions?

Reviewer #1: Yes

Reviewer #2: Yes

2. Has the statistical analysis been performed appropriately and rigorously? 

Reviewer #1: Yes

Reviewer #2: Yes

3. Have the authors made all data underlying the findings in their manuscript fully available?

Reviewer #1: Yes

Reviewer #2: Yes

4. Is the manuscript presented in an intelligible fashion and written in standard English?

Reviewer #1: Yes

Reviewer #2: Yes

5. Review Comments to the Author

Reviewer #1: The study aims are stated clearly; however, the introduction would benefit from a more detailed justification for why non-medical female university students were chosen as the target population.

Consider including references to previous studies on CKD awareness in similar populations to strengthen the rationale.

The cross-sectional design is appropriate, but additional details on sampling methodology are needed.

How were missing data handled in the analysis? Were there any exclusions due to incomplete responses?

Some sentences are lengthy and could be restructured for clarity. A thorough proofreading is recommended.

Terminology Consistency: Ensure consistent use of terms such as "CKD knowledge" versus "CKD awareness."

Standardize references and ensure all citations follow the PLOS ONE guidelines.

Reviewer #2: Thank you very much for allowing me to review this article. The study contributes usefully to understanding the awareness of chronic kidney disease among the young non-medical population in Saudi Arabia. Below, I provide recommendations for improvement.

On page 1, you mention the role of women in health education within families. Consider providing a specific citation or example to support this statement.

Page 2, kindly include a brief explanation of how the questionnaire was validated (e.g., previous studies or pilot testing) to enhance the reliability of the instrument.

Page 3, describe how missing or incomplete data were handled during the analysis. For example, did you exclude participants with incomplete answers or use statistical imputation?

Page 3, Table 1: Add percentages alongside numerical values to facilitate interpretation of demographic distributions.

Page 3, For transparency, include confidence intervals for key associations (as well as p-values) in the main text.

Page 4, Indicate whether any multicollinearity checks were performed for regression models.

Page 4 Table 2. While the data is clear, consider bolding statistically significant p-values (<0.05) to draw attention to the main findings.

Page 4, add a brief summary of the findings for each knowledge domain (general knowledge, risk factors, complications, and preventive measures) to provide a clearer overview in the text.

Page 5, Discuss the potential impact of social media and digital campaigns on CKD awareness in the target population. This is in line with the hypothesis that younger generations may access health information through these channels.

Page 6, Explain in detail why knowledge levels are higher at some universities (e.g., King Saud University). Were there specific programs or initiatives in place?

Page 6, Provide specific examples of the types of health education programs that could be implemented. For example, consider mentioning awareness campaigns and workshops or incorporating CKD topics into university curricula.

Format (if applicable): Ensure that all figures (if included) are high-resolution and appropriately labeled for clarity.

Review the manuscript for minor grammatical inconsistencies, such as unnecessary repetition of phrases.

6. PLOS authors have the option to publish the peer review history of their article (what does this mean? ). If published, this will include your full peer review and any attached files.

**Do you want your identity to be public for this peer review?** For information about this choice, including consent withdrawal, please see our Privacy Policy .

Reviewer #1: **Yes: ** Nasser M Alorfi

Reviewer #2: No

---

## [Author Response · Author response to Decision Letter 1]

27 Apr 2025

Response to Reviewer’s Comments:

Reviewer #1:

# Comment 1: The study aims are stated clearly; however, the introduction would benefit from a more detailed justification for why non-medical female university students were chosen as the target population.

Justification: Thank you for your comment. A paragraph is added in the introduction to describe this point as the following:

“Targeting non-medical female university students to assess Chronic Kidney Disease (CKD) awareness is strategic for identifying general knowledge gaps, addressing gender-specific disparities and risks, and leveraging women's roles in health promotion. Moreover, it is justified by their unique hormonal and reproductive risk factors for CKD, the opportunity to assess health behaviors and promote health literacy, their accessibility as a research population, and the chance to address potential CKD knowledge gaps and misconceptions. This approach facilitates the development of targeted educational interventions to enhance early detection and prevention efforts.”

# Comment 2: Consider including references to previous studies on CKD awareness in similar populations to strengthen the rationale.

Justification: Thank you for your suggestion. Therefore, similar previous studies were cited as the following:

“Similar study with the same target group was conducted among non-medical students in the University of Nigeria [1], Rwanda [2], Chennai [3], and in USA [4].”

# Comment 3: The cross-sectional design is appropriate, but additional details on sampling methodology are needed.

Justification: Thank you for your comment. This section of the methodology has been revised and updated as follows:

This study employed a descriptive cross-sectional survey design targeting non-medical female university students in Riyadh, Saudi Arabia. Ethical approval was obtained from the Institutional Review Board (IRB) at AlMaarefa University (Approval No. IRB23-041). Prior to participation, verbal consent was obtained from all students after explaining the study's purpose and ensuring voluntary participation. Data collection began on February to March, 2023, with the start date marking the formal enrollment of participants. The study included female students from non-medical disciplines Female medical students and all male students were excluded.

The sample size was determined using a prevalence rate of 27.8% for adequate knowledge of chronic kidney disease (CKD), based on a previous study from Jazan. Using the standard formula with Z = 1.96, P = 0.278, q = 0.65, a 95% confidence level, and a 5% margin of error, an initial sample of 305 participants was calculated. This was adjusted for a design effect of 1.5, resulting in a final sample size of 523.

A multistage sampling technique was applied. Across the 16 universities in Riyadh (comprising 11 private and 5 public institutions) four universities were randomly chosen in the study (2 publics and 2 private). Participants were proportionally distributed based on the student population in each selected institution. For public universities, King Saud University and Imam Mohammad Ibn Saud Islamic University, students were selected with a total of 334 participants equally split between the two institutions. For private universities, Prince Sultan University and Al-Yamamah University, students were selected with a total of 166 participants equally distributed between both universities.

# Comment 4: How were missing data handled in the analysis? Were there any exclusions due to incomplete responses?

Thank you for your query:

Justification: Missing data were handled by excluding incomplete responses; only fully completed questionnaires were included in the final analysis to ensure data accuracy and consistency.

# Comment 5: Some sentences are lengthy and could be restructured for clarity. A thorough proofreading is recommended.

Justification: Thank you for your comment. A thorough proofreading was conducted

# Comment 6: Terminology Consistency: Ensure consistent use of terms such as "CKD knowledge" versus "CKD awareness."

Justification: Thank you for your comment. It is done in the modified version

# Comment 7: Standardize references and ensure all citations follow the PLOS ONE guidelines.

Justification: Thank you for your comment. It is done in the modified version.

Reviewer #2: Thank you very much for allowing me to review this article. The study contributes usefully to understanding the awareness of chronic kidney disease among the young non-medical population in Saudi Arabia. Below, I provide recommendations for improvement.

# Comment 1: On page 1, you mention the role of women in health education within families. Consider providing a specific citation or example to support this statement.

Justification: Thank you for your suggestion, and accordingly, the following statement was added with citation:

“In Saudi Arabia, women serve as central figures in family health education, guiding practices related to hygiene, nutrition, and disease prevention. Cultural and religious values position them as key caregivers, with their influence further supported by national goals like Vision 2030. Their role became especially evident during the COVID-19 pandemic, as they enforced safety measures and helped manage household responses. Studies recognize Saudi women as the "first line of defense" in protecting families and promoting public health awareness [5]”

# Comment 2: Page 2, kindly include a brief explanation of how the questionnaire was validated (e.g., previous studies or pilot testing) to enhance the reliability of the instrument.

Justification: Thank you for your valuable comment, and accordingly, this part was expanded as the following:

“A structured, self-administered questionnaire—designed based on validated instruments from prior studies [1, 6-8] , was distributed as printed copies to target participants across multiple university campuses. This in-person approach ensured direct participant engagement, comprehensive responses, and efficient data collection. Data collection was conducted from September 4 to October 19, 2023, by a team of trained students from the College of Medicine at AlMaarefa University, who followed standardized protocols for accuracy and consistency. The questionnaire was first piloted among 5% of the sample (35 students), and then reviewed and assessed by 3 subject experts for its content, design, relevance, readability and comprehension. A content validity ration (CVR) was calculated for each domain and all domains reported 0.75 for knowledge domain, 0.70 for attitude domain, and 0.76 for practice domain for Cronbach alpha coefficients score. Domains with less than 0.07 were reviewed carefully and targeted questions were modified or deleted accordingly. Three questions related to the knowledge from the piloted version were found not appropriate and then not included in the final version of the questionnaire”.

The final version of the questionnaire comprised 53 questions, structured into six main sections: sociodemographic characteristics (7 questions), knowledge of CKD (12 questions), CKD risk factors related questions (11 questions), CKD complication related questions (7 questions), questions related to Active prevention of CKD (8 questions), and questions related to Source of information on chronic kidney disease (8 questions). Some questions were scored with a binary system, where incorrect or uncertain ("don't know") responses received a score of 0, while correct answers were awarded 1 point. The questions related to CKD risk factors, CKD complication, q and questions related to Active prevention utilized a five-point Likert scale (strongly disagree to strongly agree), with responses scored from 1 (least appropriate) to 5 (most appropriate). To account for unfavorable statements, scores for such items were inverted during analysis. These domains (CKD risk factors, CKD complication, q and questions related to Active prevention) had a possible score range of 11–55, 7-35, and 8-40, respectively. Criteria for correct/incorrect answers were determined through prior literature and expert validation during questionnaire development. This scoring framework ensured consistency in evaluating participants’ related status, which is aligning with established methodologies to enhance reliability and comparability of results.

# Comment 3: Page 3, describe how missing or incomplete data were handled during the analysis. For example, did you exclude participants with incomplete answers or use statistical imputation?

Justification:Thank you for your comment. It is added as:

“Missing data were handled by excluding incomplete responses; only fully completed questionnaires were included in the final analysis to ensure data accuracy and consistency.”

# Comment 4: Page 3, Table 1: Add percentages alongside numerical values to facilitate interpretation of demographic distributions.

Justification: Thank you for your comment. Done accordingly.

# Comment 5: Page 3, For transparency, include confidence intervals for key associations (as well as p-values) in the main text.

Justification: Thank you for your suggestion. I agree, confidence intervals were used accordingly when describing the key association on the regression model.

# Comment 6: Page 4, Indicate whether any multicollinearity checks were performed for regression models.

Justification: Thank you for your suggestion:

# Comment 7: Spearman's test was used to analyze the correlation between ordinal variables (academic year and other sociodemographic data) and knowledge of CKD, ensuring multicollinearity checks prior to regression analysis.

# Comment 8: Page 4 Table 2. While the data is clear, consider bolding statistically significant p-values (<0.05) to draw attention to the main findings.

Justification: I agree, and was done accordingly.

# Comment 9: Page 4, add a brief summary of the findings for each knowledge domain (general knowledge, risk factors, complications, and preventive measures) to provide a clearer overview in the text.

Justification: Thank you for your suggestion. It is added as:

“The study found that students' knowledge of CKD—across general information, risk factors, complications, and preventive measures—was significantly associated with their university and academic year. Prevention knowledge was higher in King Saud and Prince Sultan students compared to Al-Yamamah. Educational context drives awareness more than other demographics. Other sociodemographic factors did not show significant associations, indicating that institutional and academic-level differences play a key role in shaping CKD-related knowledge.”

# Comment 10: Page 5, Discuss the potential impact of social media and digital campaigns on CKD awareness in the target population. This is in line with the hypothesis that younger generations may access health information through these channels.

Justification: I agree, and therefore, this sentence was added in the text:

Digital campaigns and social media offer great promise for raising Chronic Kidney Disease (CKD) awareness among younger, who frequently use these tools for information. Consistent with the idea that younger generations turn to social media, websites, and health apps for knowledge, these tools enable targeted health education. Through interactive features, infographics, brief videos, and influencer-driven efforts, they can break down intricate medical details, making CKD risks, symptoms, and prevention strategies more approachable and compelling.

# Comment 11: Page 6, Explain in detail why knowledge levels are higher at some universities (e.g., King Saud University). Were there specific programs or initiatives in place?

Justification: Thank you for your comment. The following statement was developed for this purpose:

“The higher knowledge levels on CKD among students at King Saud University (KSU) can be linked to its competitive admission standards, which attract academically strong students [9]. The university emphasizes quality teaching, supported by experienced faculty and multimedia tools, creating an engaging learning environment [10]. Additionally, its wide range of extracurricular activities provides students with learning opportunities beyond the classroom, further enhancing their knowledge and academic growth [11].”

# Comment 12: Page 6, Provide specific examples of the types of health education programs that could be implemented. For example, consider mentioning awareness campaigns and workshops or incorporating CKD topics into university curricula.

Justification: Thank you for your suggested idea. It was implemented as the following:

“Targeted health education programs can boost CKD awareness among non-medical female students at Saudi Universities. Awareness campaigns on popular Saudi social media like Twitter and Instagram, using hashtags like “#KnowYourKidneys” with engaging content on risk factors (e.g., poor diet, inactivity) and prevention (e.g., hydration, exercise), aligned with events like World Kidney Day [1]. Interactive workshops led by medical professionals, offering screenings and tailored discussions [2], and integrating CKD topics into general university courses [3], can provide foundational knowledge. Peer-led initiatives like student-hosted “Kidney Talks” can further engage students informally [4]. These strategies address awareness gaps effectively.”

---

## [Editor Report · Decision Letter 1]

30 Apr 2025

The Awareness of Chronic Kidney Disease in Non-Medical Female University Students: A Cross-Sectional in Riyadh, Saudi Arabia

PONE-D-24-49522R1

Dear Dr. Fareed,

We’re pleased to inform you that your manuscript has been judged scientifically suitable for publication and will be formally accepted for publication once it meets all outstanding technical requirements.

Kind regards,

Jordan Llego, PhD ELM, D. Hon. Ex., PhDN, RN

Academic Editor

PLOS ONE

Additional Editor Comments (optional):

Thank you for your thoughtful and comprehensive revision of the manuscript "The Awareness of Chronic Kidney Disease in Non-Medical Female University Students: A Cross-Sectional in Riyadh, Saudi Arabia." I am pleased to inform you that after carefully considering the revised manuscript and your detailed point-by-point responses to the reviewers' comments, I have decided to accept your manuscript for publication in PLOS ONE.

The reviewers' concerns have been thoroughly addressed, and significant improvements have been made to the manuscript's clarity, methodological rigor, and scholarly value. Notably, your enhanced justification for the target population, expanded methodological description, and improved reporting of statistical analyses—including confidence intervals and multicollinearity checks—have strengthened the study considerably.

But please revise the title to The Awareness of Chronic Kidney Disease in Non-Medical Female University Students: A Cross-Sectional Study in Riyadh, Saudi Arabia.

Your work presents valuable insights into an underexplored chronic kidney disease awareness research demographic. The findings provide a solid foundation for future health education initiatives and are well-aligned with the journal's mission to promote scientific knowledge with real-world impact.

Congratulations once again. We look forward to publishing and sharing your article with the global research community.
---

## [Editor Report · Acceptance letter]

PONE-D-24-49522R1

PLOS ONE

Dear Dr. Fareed,

I'm pleased to inform you that your manuscript has been deemed suitable for publication in PLOS ONE. Congratulations! Your manuscript is now being handed over to our production team.

Kind regards,

on behalf of

Dr. Jordan Llego

Academic Editor

PLOS ONE